# Competitiveness of Xinjiang's mutton industry based on diamond model

**Ablat Dawut****, Yingjie Tian***

School of Economics and Management, University of Chinese Academy of Sciences, Beijing, China

* tyj@ucas.ac.cn

**Data Availability Statement:** All relevant data are within the manuscript.

**Funding:** The authors received no specific funding for this work.

## Abstract

In recent years, Xinjiang mutton production has experienced a growth trend; however, it cannot meet the new consumer demand. Based on Michael Porter's "diamond model," this study presents a case study on the Xinjiang mutton industry in China and establishes an index system for the competitiveness of the industry. The competitiveness of the mutton industry is analyzed quantitatively via correlation analysis and principal component analysis by investigating the relevant data of 10 Chinese provinces topping in mutton production. On account of the related elements of the diamond model, a qualitative analysis is also performed. The quantitative analysis shows that among the 10 provinces (regions) topping in mutton production in China, Inner Mongolia wins in competitiveness, followed by Xinjiang, and Henan ranks at the bottom. The qualitative analysis shows that the Xinjiang mutton industry is inferior in three main factors compared to its competitors, and these are the production factors related to and supporting the industries and the enterprise strategies. Xinjiang performs moderately in terms of the auxiliary government factors. However, Xinjiang is in an advantageous position with respect to the main factors of demand conditions and auxiliary elements of opportunities. Given the existing problems, this study discusses the main reasons for the lack of competitiveness of the Xinjiang mutton industry. It also puts forward some strategic suggestions to enhance the competitiveness of the Xinjiang mutton industry based on the six elements of diamond model.

## Introduction

The mutton industry plays a vital role in the animal husbandry economy. It is also an indispensable part to measure the competitiveness of regional animal husbandry and is an important indicator of animal husbandry. At present, to protect the ecological environment, China further established the grassland ecological protection subsidy and reward mechanism in 2011 [1]. Some policies have been implemented, such as banning grazing, balancing grass and livestock, and returning grazing to grassland [2]. The policy implementation will inevitably bring about the transformation of the mode of animal husbandry production [3]. Consequently, the shrinking of the natural grazing area has posed certain challenges for the traditional mutton industry, leading to important changes in the production mode and the variables of industrial competitiveness. The mutton industry is facing new development opportunities, with the

**Competing interests:** The authors have declared that no competing interests exist.

transformation of the production mode of the mutton industry and the enhancement of its connection with other related industries.

China is competent in both mutton production and consumption [4]. Since the reform and opening up, the mutton industry in China has developed rapidly, and its output has greatly increased, with mutton production topping in the world for years [5]. In 2019, the national mutton output was 4.8752 million tons, of which the Xinjiang output was 603,200 tons, accounting for 12.37% of the total and ranking second in China. Driven by the economic development and advancements in the tourism industry, the Xinjiang mutton industry has seen steady development and has become an important pillar in animal husbandry. Nevertheless, certain problems in the industry still exist, weighing down on its competitive advantage. Based on the diamond model of Michael Porter, this study first constructs an index system for the competitiveness of the mutton industry. It then delves into a comprehensive analysis of the competitiveness to explain the current situation of the mutton industry in Xinjiang. Finally, this study proposes some suggestions for enhancing the industrial competitiveness of the Xinjiang mutton industry.

By the end of 2019, Xinjiang had a total population of 25.23 million. Moreover, the total output of meat in Xinjiang was 1.5373 million tons, among which the mutton accounted for 0.6032 million tons or 37.88% of the total meat output. Mutton is prioritized in the meat choice and consumption in Xinjiang [6], and it amounts to a rather high proportion of various ethnic groups' diet, thus making the consumption of mutton irreplaceable in the region. In 2019, Chinese residents' per capita meat consumption was 26.7 kg, including 3.76 kg of mutton. In Xinjiang, the per capita meat consumption was 21.6 kg, of which mutton consumption was 13.9 kg, ranking first in China. The mutton industry is dominant in the development of animal husbandry in the Xinjiang autonomous region [7]. Hence, practically, the development of the mutton industry must be sped up to optimize the industrial structure of agriculture and animal husbandry by ensuring the market supply, promoting the efficiency, and increasing the income of farmers and herders. *Note*: The data are sourced from *China Statistical Yearbook* and *Xinjiang Statistical Yearbook*.

As indicated in Fig 1, between 2006 and 2009, the mutton production ranged from 669,900 tons to 437,700 tons, showing a 33.2% decrease. From 2009 to 2019, stagnant growth was

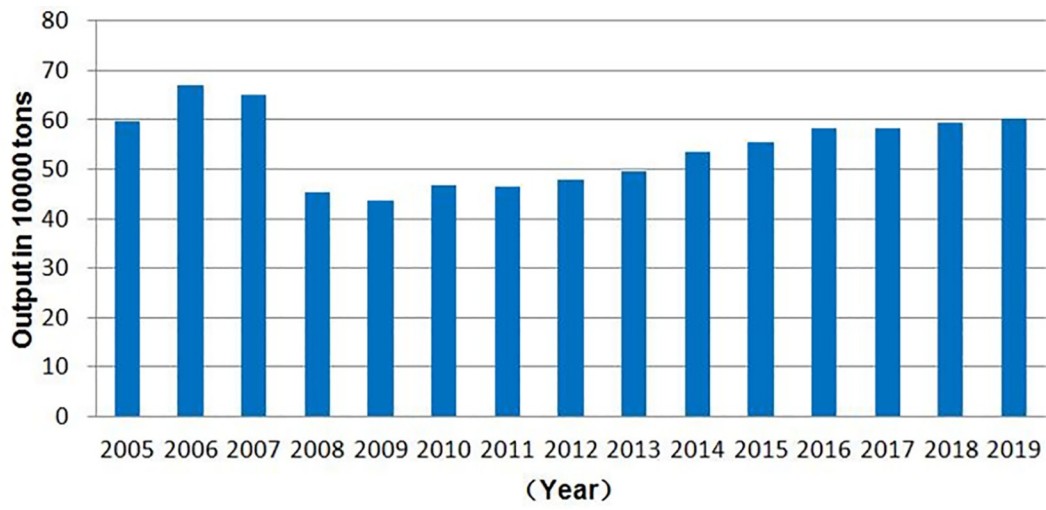

**Fig 1. Mutton production in Xinjiang, 2003–2019 (10,000 tons).** *Note*: The data sources are arranged according to *Xinjiang Statistical Yearbook*.

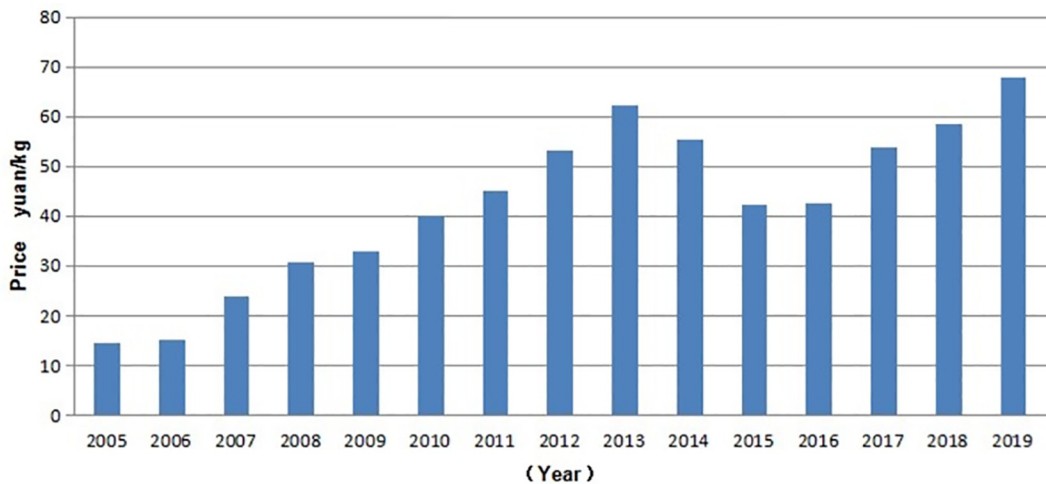

**Fig 2. Xinjiang mutton price, 2003–2019 (yuan/kg).** *Note*: The data source is arranged according to the *Yearbook of China Animal Husbandry and Veterinary Medicine* (2003–2017, market price in December each year). Due to the lack of animal products and feed prices in December 2003, the price of mutton in November 2003 was replaced.

maintained. The production remained almost the same in 2005 and 2019. In summary, mutton production in Xinjiang saw a wide range of changes, such as rapid growth, downhill decline, and slow growth, in the recent 15 years ending in 2019.

As shown in Fig 2, between 2005 and 2013, the price of mutton rose sharply to 62.23 yuan. Between 2013 and 2016, the mutton prices showed a downward trend and gradually returned to the level of where they were three years before. In 2017, a growth trend started again, and in 2019, the highest price of Xinjiang mutton (67.88 yuan) was reached. In short, the price of mutton in Xinjiang has maintained an increasing trend in the last 15 years, during which period, only one downward trend existed, and the average annual growth was 24.33%. The highest price was recorded in 2019, which was 365% higher than the price in 2005.

In 2019, the permanent population of Xinjiang was 25.23 million, showing a 25.5% increase compared to 20.1035 million in 2005, with an average annual growth of 1.7%. In the same year, the annual tourist visits in Xinjiang were 213.2954 million, with a 1,323.76% increase compared to the 14.9811 million in 2005. Moreover, it had an average annual growth of 88.25%. The CPI in 2019 was 695.5 yuan, which was 48.42% higher than that in 2005 (i.e., 468.6 yuan). Hence, the growth of population, tourists, and CPI could be the main factors increasing the price of mutton. *Note*: The data are sourced and sorted according to *Xinjiang Statistical Yearbook*.

In 2019, Xinjiang residents' per capita mutton consumption expenditure was 17,397 yuan, showing a 180.2% increase compared to 6,208 yuan in 2005. As seen from the individual consumption levels in Fig 3, the consumption of mutton had increased; this could be due to the increase in the population and the number of tourists. According to the statistics of the Institute of Animal Health Supervision of Xinjiang Uygur Autonomous Region, 1.47 million live sheep were transferred from other provinces to Xinjiang in 2019, and 256,000 were transferred from Xinjiang to other provinces. The transfer-in quantity is much higher than the transfer-out quantity. Xinjiang local mutton cannot meet the new consumer demand.

As shown in Fig 3, from 2005 to 2017, the consumption of mutton in Xinjiang maintained a slow growth trend, with an increase of 32.4% and an average annual growth of 2.1%. In 2019, the per capita mutton consumption of Xinjiang residents was 13.9 kg. Although 25.23 million

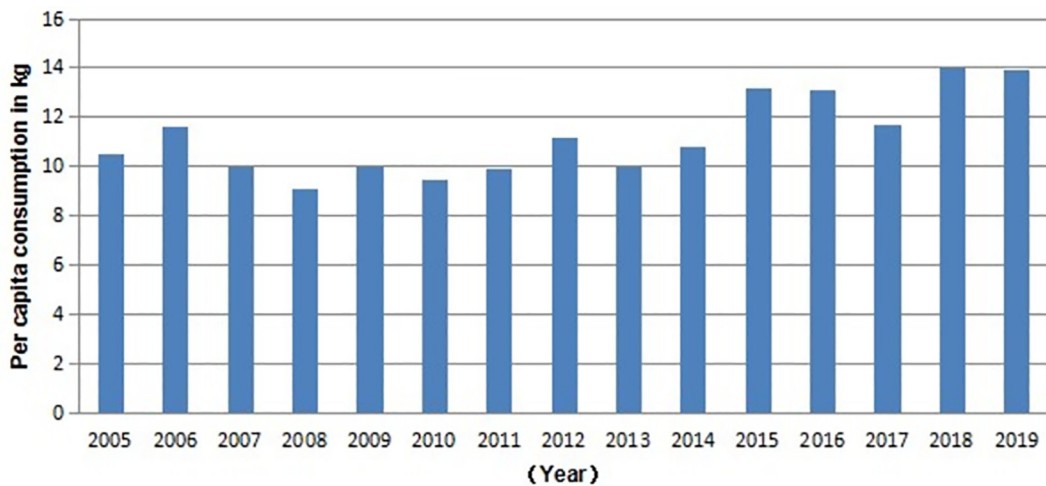

**Fig 3. Per capita consumption of mutton in Xinjiang, 2003–2017 (kg).** *Note*: The data sources are arranged according to *Xinjiang Statistical Yearbook*.

permanent residents were in Xinjiang at the end of 2019, the total consumption of mutton of Xinjiang residents was only 350,700 tons. However, the total output of mutton in Xinjiang was 603,200 tons, with a surplus of 252,500 tons. If the total amount of mutton consumption is accurately calculated, we can only account for the remaining 252,500 tons of mutton consumption to the tourists. Xinjiang received 213.2954 million tourists throughout the year in 2019, and the tourists' per capita consumption of mutton was 1.18 kg. According to the abovementioned analysis and calculation, the total consumption of mutton in Xinjiang was close to its total output of mutton in the given year.

In recent years, Xinjiang mutton production has gradually increased, but meeting the demands of local market is difficult. Many problems still exist in Xinjiang mutton industry, and the competitive advantage is difficult to highlight. Although many experts and scholars used Michael Porter's diamond model to study industrial competitiveness, most of them used qualitative analysis, whereas only a few scholars utilized the diamond model for quantitative and qualitative analyses of industrial competitiveness. The diamond model can be used as an analysis tool for a single industry [8]. Therefore, based on Michael Porter's diamond model, this study constructs the index system of mutton industry competitiveness, comprehensively analyzes the competitiveness of mutton industry, clarifies the current situation of Mutton Industry in Xinjiang, and puts forward suggestions for improving the competitiveness of the mutton industry.

## Methods and data sources

### Methods

The diamond model (Fig 4) is a comprehensive and systematic analysis framework of industrial competitiveness proposed by Michael Porter in his masterpiece, *The Competitive Advantage of Nations*, in 1990. The model is composed of four main factors and two auxiliary factors. The six factors influencing and impacting each other form an integral system and pose a joint effect on industrial competitiveness [9]. Researchers, such as Fan and Liang [10, 11], built the evaluation system of industrial competitiveness from the four main elements, according to the diamond model. Similarly, Ding and Zhao [12] constructed an analytical framework for the

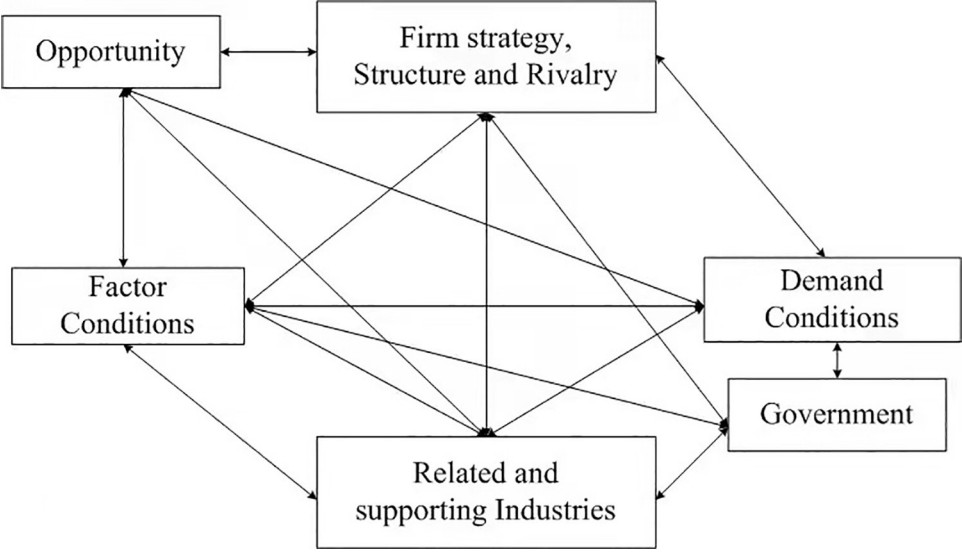

**Fig 4. Michael Porter's "diamond model".**

international competitiveness of Chinese agricultural products. Gao [13] analyzed the influencing factors of agricultural competitiveness in Hainan from the six main factors using correlation analysis. Meanwhile, Zhou and Li [14, 15] selected five first-class indicators of the four major elements and one auxiliary element, the "government." However, these scholars did not investigate the two auxiliary factors, or they only studied the auxiliary factors of the "government." They believed that the two auxiliary factors of "opportunity" and "government" were realized through the first four main factors.

Different scholars have made in-depth analyses of the influencing factors of the mutton industry's international competitiveness by utilizing various econometric models. For instance, Cui et al. [16] applied the gravity model to realize an empirical analysis of the main factors affecting the competitiveness of Chinese mutton on the global market. Duan et al. [17] applied the equilibrium model to analyze the impact of different proportions of export volumes on the international competitiveness of Chinese mutton. Using the two-step method of DID and Heckman, Wang et al. [18] studied the influence of the following three aspects on the international competitiveness of the Chinese mutton industry: negotiation, increasing imports, and evaluation of the related impacted industries. Some major research methods, including the diamond model, applied to agricultural competitiveness analysis are the fuzzy comprehensive evaluation method [19], SWOT analysis method [20], principal component analysis, data envelopment analysis method [21], cluster analysis method [22], factor analysis method [23], and analytic hierarchy process [24]. In addition, some scholars have conducted studies on how to improve agricultural competitiveness, and they achieved great success by using the foreign national diamond model, theories of comparative advantage, factor endowment, demand elasticity, innovation, natural monopoly, and differentiation, and other perspectives [25]. Although studies on the qualitative analysis of industrial competitiveness using the diamond model are plenty, only a few studies have conducted the quantitative analysis of industrial competitiveness. The application of diamond model in the study of mutton industry competitiveness has applicability.

The main feature of this paper is to conduct a quantitative and qualitative analyses on the competitiveness of the Xinjiang mutton industry based on the diamond model. The primary quantitative analysis method is to build the competitiveness evaluation index of the mutton industry, using the original data of mutton production in the top 10 provinces in China. Moreover, correlation analysis and principal component analysis are conducted to obtain the comprehensive competitiveness evaluation data of 10 provinces. The main qualitative analysis method is to analyze the competitiveness of Xinjiang mutton industry based on the four main elements and two auxiliary elements of the diamond model.

## Quantitative analysis

The key to the comprehensive evaluation of mutton industry competitiveness lies in the scientific selection of indicators and the establishment of an index system and its operability. In this paper, we should make full use of the main and auxiliary elements of the diamond model, to reveal not only the main influencing factors of the mutton industry's competitiveness but also the auxiliary influencing factors. Moreover, this method is used to reflect the importance, purpose, and content of evaluation, and prevent one sidedness. The indicators should be interrelated and coordinated to reflect the integrity and avoid strong linear relationship. Moreover, the index system should use the real statistical data to analyze and achieve the comparability, rationality, and objectivity of the results.

According to the production factors, demand conditions, performance of related and supporting industries, strategy, structure, and competition of enterprises and government of diamond model, this study determines the competitiveness evaluation indexes of the mutton industry. Moreover, 5 first-class indexes, 8 second-class indexes, and 16 third-class indexes were selected to evaluate the competitiveness of the mutton industry. This study does not consider the "opportunity" auxiliary factors as these factors can play a role through other factors and are difficult to quantify (Table 1).

**Table 1. The evaluation index of mutton industry competitiveness.**

| First-class indicators | Second-class indicators | Third-class indicators | Variables |
|---|---|---|---|
| Productive factors | Primary productive factors | Number of raised sheep (10000) | X1 |
| | | Mutton production (10000 tons) | X2 |
| | Advanced productive factors | Animal husbandry technical institutions (individual) | X3 |
| | | Animal husbandry Technician (person) | X4 |
| Demand conditions | Local consumption | Total population (10000) | X5 |
| | | Consumption level of residents (yuan) | X6 |
| | Tourist market | Number of tourists (10000) | X7 |
| | | Tourism income (100 million yuan) | X8 |
| Performance of related industries and supporting industries | Cultivation scale | Sheep stud farm (individual) | X9 |
| | | Large scale sheep farm (more than 500 sheep) | X10 |
| | Catering trade | Corporate enterprise of catering trade (individual) | X11 |
| | | Number of employees in catering enterprises (person) | X12 |
| Strategy, structure and competitors of enterprises | Competitiveness of animal husbandry | Output value of animal husbandry (10000 yuan) | X13 |
| | | Mutton brands favored by consumers (individual) | X14 |
| | | Large scale sheep farm with more than 500 annual column quantity (household) | X15 |
| Government | Governmental investment | Fixed assets investment in agriculture, forestry, animal husbandry and fishery (100 million yuan) | X16 |

**Table 2. The original data of the mutton index system.**

| Third-class indicators | Inner Mongolia | Xinjiang | Shandong | Hebei | Sichuan | Henan | Gansu | Yunnan | Anhui | Hunan |
|---|---|---|---|---|---|---|---|---|---|---|
| Number of raised sheep (10000) | 6112 | 4318 | 1754 | 1228 | 1599 | 1682 | 1840 | 1240 | 505 | 662 |
| Mutton production (10000 tons) | 104 | 58 | 36 | 30 | 27 | 26 | 23 | 18 | 17 | 15 |
| Animal husbandry technical institutions (Individual) | 1202 | 1239 | 1664 | 1665 | 4271 | 1590 | 1509 | 1587 | 1293 | 2120 |
| Animal husbandry technician (Person) | 12306 | 14490 | 11186 | 10641 | 21471 | 13418 | 10702 | 9352 | 5549 | 15404 |
| Total population (10000) | 2529 | 2445 | 10006 | 7520 | 8302 | 9559 | 2626 | 4801 | 6255 | 6860 |
| Consumption level of residents (yuan) | 23909 | 16736 | 28353 | 15893 | 17920 | 17842 | 14203 | 15831 | 17141 | 19418 |
| Number of tourists (10000) | 11646 | 10726 | 78000 | 57000 | 57000 | 66511 | 23905 | 57300 | 63100 | 67323 |
| Tourism income (100 million yuan) | 3440 | 1822 | 9200 | 6141 | 8923 | 6751 | 1540 | 6922 | 6197 | 7173 |
| Sheep stud farm (individual) | 439 | 104 | 42 | 51 | 108 | 48 | 181 | 132 | 83 | 32 |
| Large scale sheep farm (more than 500 sheep) | 3484 | 2639 | 10144 | 7581 | 4420 | 10038 | 2019 | 3535 | 6313 | 6255 |
| Corporate enterprise of catering trade (individual) | 326 | 97 | 1767 | 429 | 1474 | 1235 | 362 | 536 | 1213 | 930 |
| Number of employees in catering enterprises (person) | 26404 | 9577 | 121848 | 32554 | 96640 | 67165 | 22204 | 31165 | 76554 | 60737 |
| Output value of animal husbandry (10000 yuan) | 1201 | 749 | 2501 | 1736 | 2200 | 2369 | 309 | 1290 | 1322 | 1506 |
| Mutton brands favored by consumers (individual) | 52 | 8 | 12 | 0 | 0 | 1 | 2 | 0 | 0 | 0 |
| Large scale sheep farm with more than 500 annual column quantity (household) | 10741 | 4983 | 2325 | 3133 | 795 | 1518 | 1594 | 265 | 1498 | 374 |
| Fixed assets investment in agriculture, forestry, animal husbandry, and fishery (100 million yuan) | 1153 | 639 | 1610 | 1873 | 1492 | 2676 | 407 | 1268 | 896 | 1579 |

## Data sources

To further explain the competitiveness of the Xinjiang mutton industry, this study selected the top 10 provinces and regions of China in mutton output in 2017. Moreover, the relevant data are collected from the *China Statistical Yearbook*, the *China Animal Husbandry* and *Veterinary Medicine Yearbook*, and the *Statistical Yearbooks* of the 10 provinces and regions, such as Inner Mongolia (Table 2).

## Data analysis

A total of 16 variables are related to mutton industry competitiveness, but the sample data consist of 10 groups (provinces). The variables are greater in number than the samples; hence, conducting the principal component analysis directly is difficult. Consequently, a correlation analysis must be conducted to eliminate certain variables with an overly high correlation with other variables.

## Correlation analysis

First, we use the package of SciPy in Python to check the significance of the Pearson correlation coefficient one by one.

*H0*. *No significant correlation exists between X and y.*

*Ha*. *A significant correlation exists between X and y.*

We set the threshold of P-value as 0.05. If it is less than 0.05, we reject the original hypothesis and conclude that a significant linear correlation exists between X and y. On the contrary, we accept the original hypothesis. The P-value of the production factor's own sheep-feeding quantity and mutton yield is 1; thus, we accept the original hypothesis but do not consider it. From the array, we find that the P-value of each three-level index in the first-level index is

close to 1, so we do not consider it. Using Python software for correlation analysis, the 16 variables are correlated to establish a correlation chart. The number in the figure is the Pearson correlation coefficient between the two variables, and the value is between +1 and −1. A positive number indicates a positive correlation between the two variables, and a negative number indicates a negative correlation between the two variables. Suppose we set the threshold of the correlation coefficient to 0.8. In that case, we can obtain the following characteristics: mutton output (10,000 tons), annual sales of more than 500 farms (households), tourism income (100 million yuan), number of mutton brands preferred by consumers, number of tourists (10,000 person times), output value of animal husbandry (10,000 yuan), sheep-breeding farms (farms), sheep breeding farms (more than 500), and the number of corporate enterprises and employees in the catering industry. In the first-level indicators, we hope to adjust the production factors, demand conditions, industrial support, and government support, and finally achieve the role of improving the output value of mutton enterprises. We did not study the indicators excluded by significance analysis. The total population (0.93), tourism income (0.88), and the number of employees in the catering industry (0.8) correlate positively with the output value of animal husbandry. From this, we can determine that stimulating tourism and catering can help improve the output value of animal husbandry. The production of mutton (0.94) and the production of sheep farms (0.88) were helpful to increase the number of mutton brands preferred by consumers.

Python software was used in the correlation analysis, and two or more variables with correlation were analyzed. The correlation relationship between every 2 of the 16 variables was established, and the correlation chart was made (see Fig 5). The darker the color, the greater the correlation between the two variables. Red indicates positive correlation, and green indicates negative correlation).

Fig 5 reveals the close correlation between sheep-feeding quantity and mutton production, annual slaughter scale and market quantity. Similarly, the correlation between mutton production and mutton brand, annual marketing scale and market quantity is close. Moreover, the total population is closely related to the number of tourists, tourism income, sheep-breeding scale, output value of animal husbandry, catering enterprises, and the number of catering employees. Further, the correlation between the number of corporate enterprises and the number of catering employees and that between mutton brand and annual market size is close to 1. Therefore, a more intuitive competitiveness index of the mutton industry should be considered comprehensively. In this study, the number of tourists, tourism income, the number of catering enterprises and employees, the sheep-feeding quantity, the mutton brand, and the number of annual markets were excluded.

## Principal component analysis

Based on the results of correlation analysis, SPSS software was used to conduct principal component analysis on the remaining nine indicators: mutton production (in 10,000 tons), animal husbandry technical institutions, animal husbandry technicians, total population (in 10,000 s), consumption level of residents (in yuan), sheep stud farms (individual), large-scale sheep farms (more than 500), output value of animal husbandry (in 10,000 yuan), and fixed assets investment in agriculture, forestry, animal husbandry, and fishery (in 100 million yuan) (see Fig 6).

For the principal component analysis, the variables whose eigenvalues are more than 1 are to be kept, and it should also be guaranteed that those principal components with gentle decreasing slopes of eigenvalue are kept. Consequently, four principal components are retained in the actual principal component analysis (the cumulative principal component

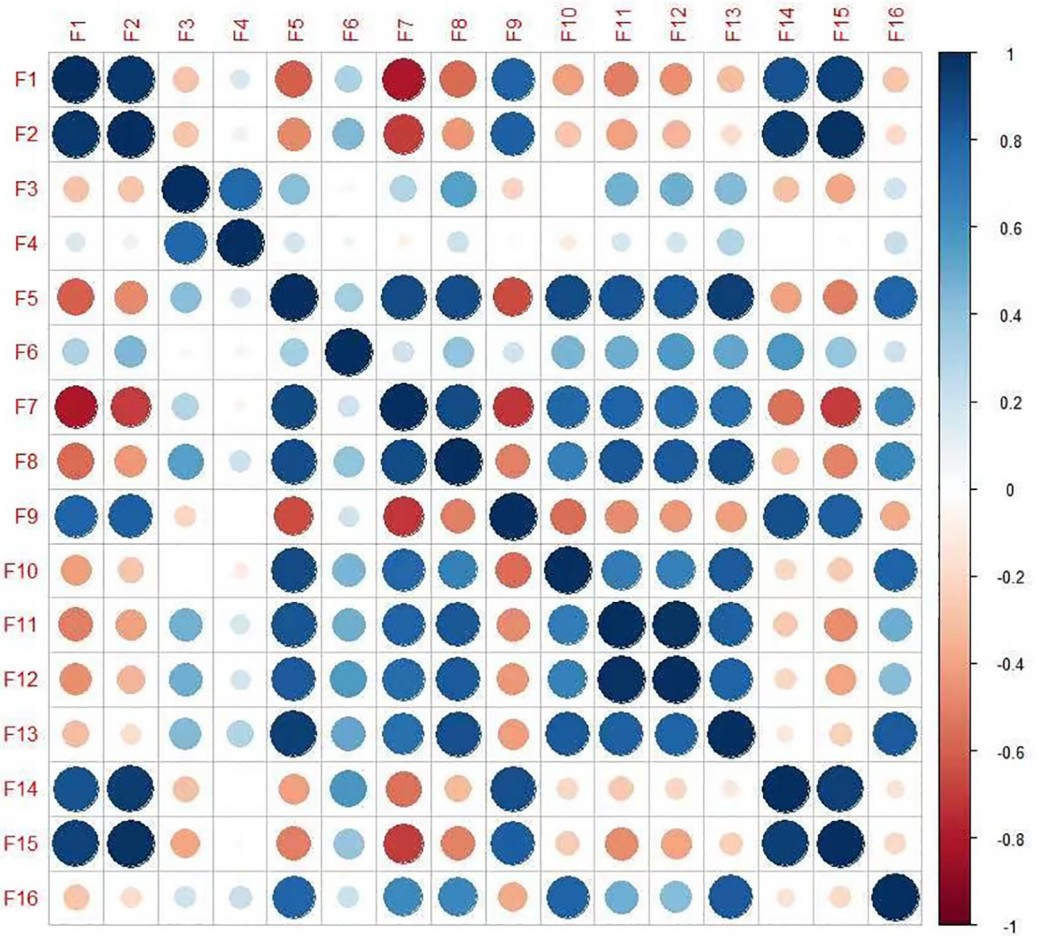

**Fig 5. Correlation among variables of mutton industry competitiveness.** *Note*: **F1:** Number of Raised Sheep, **F2:** Mutton Production, **F3:** Animal Husbandry Technical Institutions, **F4:** Animal Husbandry Technician, **F5:** Total Population, **F6:** Consumption Level of Residents, **F7:** Number of Tourists, **F8:** Tourism Income, **F9:** Sheep Stud Farm, **F10:** Large-Scale Sheep Farm, **F11:** Corporate Enterprise of Catering Trade, **F12:** Number of Employees in Catering Enterprises, **F13:** Output Value of Animal Husbandry, **F14:** Mutton Brands Favored by Consumers, **F15:** Large-Scale Sheep Farm with More than 500 Annual Column Quantities, **F16:** Fixed Assets Investment in Agriculture, Forestry, Animal Husbandry and Fishery.

decomposition rate is 95.6%). The results of the principal component analysis for each variable and each province are shown in Tables 3 and 4, respectively (see Tables 3 and 4).

Comprehensive competition index. Using the first four principal components and their weights, we calculate the corresponding comprehensive competition index of each province (Table 5).

## Qualitative analysis

**Index selection.** On the basis of the production factors of the diamond model, this study conducts a qualitative analysis on the competitiveness of the mutton industry in Xinjiang to realize which four of the main factors (i.e., demand conditions, related and supporting industries, industrial strategies, structure, and horizontal competition) and the two auxiliary factors (opportunity and government) are employed.

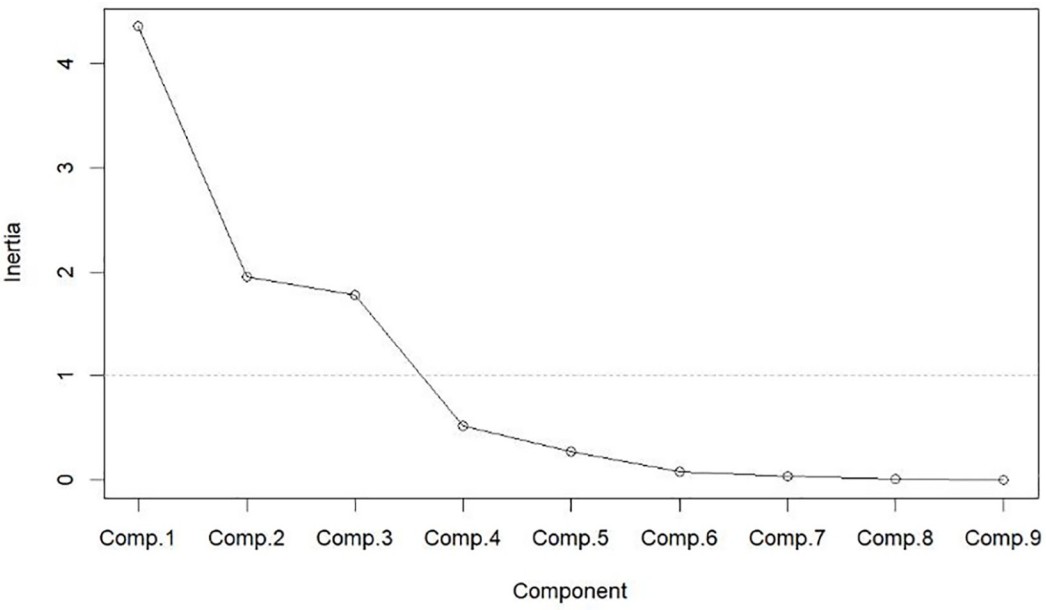

**Fig 6. Gravel map of principal component analysis.**

**Production factors (Factor conditions).** Production factors include both primary and advanced production factors. Primary production factors refer to natural resources, whereas advanced production factors refer to scientific research and human resources.

*Natural resources.* Xinjiang has 860 million mu of natural grassland (about 20% of the total grassland area in China), of which 720 million mu of natural grassland and 130 pastures can be used, providing a basis for the development of animal husbandry [26]. In 2011, the state began implementing the grassland ecological protection subsidy award and compensation mechanism in eight grassland provinces [27]. With the implementation of the grassland ecological protection subsidy award, the contradictions between the economic interests of the herders and grassland ecological protection have become prominent, and the traditional living strategies in pastoral areas have been challenged [28]. Xinjiang has more than 20 local sheep breeds. They have strong adaptability in arid and cold areas, and can tolerate roughage, but their reproductive rate is low. That is, Xinjiang sheep suffer from a low birth rate, which is a key factor limiting the development of the mutton/sheep industry [29].

**Table 3. Principal component analysis results corresponding to each variable.**

| Variable | Comp.1 | Comp.2 | Comp.3 | Comp.4 |
|---|---|---|---|---|
| Mutton production | 0.22 | 0.59 | 0.13 | 0.22 |
| Animal husbandry technical institutions | -0.20 | -0.16 | 0.62 | -0.19 |
| Animal husbandry technicians | -0.13 | 0.03 | 0.69 | 0.10 |
| Total population | -0.47 | 0.00 | -0.04 | -0.10 |
| Consumption level of residents | -0.15 | 0.59 | -0.04 | -0.62 |
| Sheep stud farms | 0.32 | 0.43 | 0.15 | 0.24 |
| Large-scale sheep farms | -0.42 | 0.15 | -0.30 | 0.04 |
| Output value of animal husbandry | -0.45 | 0.20 | 0.06 | 0.01 |
| Fixed asset investment in agriculture, forestry, animal husbandry, and fishery | -0.40 | 0.13 | -0.04 | 0.68 |

**Table 4. Principal component analysis results and corresponding weights.**

| Province | Comp.1 | Comp.2 | Comp.3 | Comp.4 |
|---|---|---|---|---|
| Inner Mongolia | 2.73 | 3.39 | 0.52 | 0.46 |
| Xinjiang | 2.37 | -0.27 | 0.38 | -0.07 |
| Shandong | -2.69 | 1.74 | -0.98 | -1.41 |
| Hebei | -1.07 | -0.5 | -0.76 | 0.77 |
| Sichuan | -1.7 | -0.6 | 3.44 | -0.27 |
| Henan | -2.94 | 0.21 | -0.66 | 1.36 |
| Gansu | 2.95 | -1.45 | -0.08 | -0.25 |
| Yunnan | 0.9 | -0.97 | -0.52 | 0.2 |
| Anwei | 0.44 | -0.87 | -1.76 | -0.55 |
| Hunan | -0.98 | -0.67 | 0.41 | -0.25 |
| Weight | 0.48 | 0.22 | 0.20 | 0.06 |

*Scientific research and human factors.* Animal husbandry research institutions possess large teams but not strong ones [30]. In 2018, the number of major professional and technical personnel of enterprises and institutions in Xinjiang was 460,465, including 18,343 agricultural technicians, accounting for 3.98% of the total. At the autonomous region level, there are 302 major scientific and technological achievements, including 2 national and 144 regional scientific and technological achievements.

**Demand conditions.** Kotler [31] believed that "word-of-mouth-influence" has a great impact on the purchasing behavior of consumers. Focusing on the quality, experts adopt different methods to identify Xinjiang mutton. The protein content of Kazakh sheep is 20.81%, whereas its nonessential amino acid content and essential amino acid content are 220.62 and 86.69 mg/g, respectively. These three contents are higher than those of the market samples. The amounts of essential and nonessential amino acids in Kazakh mutton adequately meet the requirements of Food and Agriculture Organization of the United Nations (FAO) and World Health Organization (WHO) for protein. Meanwhile, the amounts of protein, amino acids, and unsaturated fat in Kazakh mutton are at a high level of 300, indicating a high nutritional value and richness in volatile flavor compounds [32]. The meat of Hotan, Karakul, Kirgiz, and Duolang sheep is rich in iron and zinc; it is a good food source for this kind of mineral element [33]. The total amino acid contents per 100 g of Turpan black sheep and Altay sheep are 18.57 g and 17.93 g, respectively. The amounts of glutamic acid and asparagus amino acid are abundant, which are the main components of amino acids in food and the reason for the delicious taste [34].

**Case: Attractive mutton kebabs and baked Nang in Xinjiang.** For Xinjiang specialty dishes, the roasted mutton kebabs and baked Nang of Xinjiang are extremely popular. The roasted mutton kebabs are so enticing because of the excellent quality of the Xinjiang mutton. In particular, the sheep growing up in the Altay and Yili grassland feed on fresh, green grass and gurgling streams. They are never artificially fattened, which makes their meat tender and free of smell. Nowadays, the roasted mutton kebabs have made their name out of Xinjiang and become popular in China [35].

**Table 5. The ranking of comprehensive competitiveness.**

| Province | Inner Mongolia | Xinjiang | Gansu | Yunnan | Sichuan | Anhui | Hunan | Hebei | Shandong | Henan |
|---|---|---|---|---|---|---|---|---|---|---|
| Comprehensive competitiveness | 2.28 | 1.21 | 1.13 | 0.14 | -0.3 | -0.37 | -0.58 | -0.76 | -1.25 | -1.49 |

**Table 6. The number of large-scale sheep farms (household) in Xinjiang.**

| Annual column quantity (sheep) | 1~29 | 30~99 | 100~499 | 500~999 | Over 1000 | Total (household) |
|---|---|---|---|---|---|---|
| Farm (household) | 1128832 | 237449 | 41658 | 4790 | 491 | 1413220 |

*Note*: Data are sourced from *China Animal Husbandry and Veterinary Yearbook*: *2018*.

The development potential of the mutton industry in China is huge. Compared to pork, beef, and poultry, mutton production is low, and the prices are high. In 2019, the total output of mutton in China was 4.8752 million tons, accounting for 6.28% of the total output of meat (77.5878 million tons). Moreover, in the meat consumption structure, the proportion of mutton consumption is low. The Dietary Guidelines for Chinese Residents (2016) pointed out that in the next decade, the herbivorous animal husbandry will grow rapidly, and mutton production is expected to grow steadily, with an average annual growth rate of 2.4%. By 2026, national mutton production will reach 5.8 million tons, the consumption will continue to increase, and high-quality regional products will be favored [36].

**Performance of related and supporting industries.** The multiple related and supporting industries of mutton production include forage resources, veterinary drug production, slaughtering, processing, packaging, cold chain, transportation, marketing, and catering. In 2019, Xinjiang marked a grain-planting area of 33.0542 million mu, including 15.9239 million mu of wheat and 14.958 million mu of corn, for guaranteeing the development of the sheep-raising industry. Fifty-nine enterprises in Xinjiang have been listed in the list of state key leading enterprises in agricultural industrialization, one of which involves mutton slaughtering and processing.

**Strategies, structures, and competitors of the enterprises.** Among the top 100 agricultural products processing enterprises in China in 2019, none is involved in animal husbandry in Xinjiang. In 2020, among the 1,120 national key leading enterprises in agricultural industrialization, 43 enterprises are in Xinjiang, including 8 enterprises involved in animal husbandry. However, none of them are involved in the mutton industry. The mutton sheep industry in Xinjiang has not yet eliminated the traditional production mode used for scattered feeding or stocking of thousands of households. This mode cannot produce mutton in a balanced batch and cannot effectively participate in the market competition. Moreover, it is not conducive to the use of advanced supporting technology and equipment to improve the yield and quality of mutton, which seriously restricts mutton industry's further development.

As shown in Table 6, large-scale farms/households (more than 1,000 sheep) account for only 0.03% of the total households/farms; meanwhile, farms (households) with 500–999 sheep account for 0.34%, and farms with 1–29 sheep account for 79.9%.

As indicated in Table 7, from 2009 to 2018, 11 mutton brands in Inner Mongolia were included 52 times in the top 10 mutton brands favored by consumers through the web portals of "meat industry media alliance." "Small sheep," "Welllamb," and "Caoyuanxingfa" were shortlisted for 10 consecutive years. However, only three mutton brands from Xinjiang were listed in the top 10 favorite brands eight times, despite the fact that Xinjiang ranked third in terms of grassland area and sheep stock, and its mutton output ranked second in China.

**Opportunity.** The Xinjiang mutton industry has opened the following opportunities. First, the initiative of "belt and road" has liberated fresh new driving forces to the development of the tourism industry in Xinjiang. Despite the downward economic pressure, the Xinjiang tourism industry has bucked the trend and developed rapidly, becoming one of the main forces of economic growth under the "new normal" situation [37]. Second, in 2019, Xinjiang received 213.2954 million tourists, a 42% increase compared with that in 2018. Moreover, the

**Table 7. The top 10 mutton brands favored by Chinese consumers (Inner Mongolia and Xinjiang), 2009–2018.**

| Province | 2009 | 2010 | 2011 | 2012 | 2013 | 2014 | 2015 | 2016 | 2017 | 2018 |
|---|---|---|---|---|---|---|---|---|---|---|
| **Inner Mongolia** | Little Sheep | Little Sheep | Little Sheep | Little Sheep | Little Sheep | Little Sheep | Little Sheep | Little Sheep | Little Sheep | Little Sheep |
| | Wellamb | Wellamb | Wellamb | Wellamb | Wellamb | Wellamb | Wellamb | Wellamb | Wellamb | Wellamb |
| | Caoyuan xingfa | Caoyuan xingfa | Caoyuan xingfa | Caoyuan xingfa | Caoyuan xingfa | Caoyuan xingfa | Caoyuan xingfa | Caoyuan xingfa | Caoyuan xingfa | Caoyuan xingfa |
| | Sunite | Sunite | Sunite | Sunite | Sunite | Sunite | Sunite | Sunite | Sunite | Eerdun |
| | | | Mengdu | Qinlv | | Grassland Hongbao | Mengdu | Mongolian Sheep | Mongolian Sheep | Mongolian Sheep |
| | | | | | | | | | Mengdu | |
| | Ujimqin | | Ujimqin | | Ujimqin | Ujimqin | | | Grassland Hongbao | Mengdu |
| **Xinjiang** | Hualing | Bakouxiang | | Bakouxiang | Bakouxiang | Altai Grassland | Bakouxiang | Bakouxiang | | |

*Note*: The source of information is sorted out according to China's meat industry website (http://www.Chinameat.cn/).

total tourism consumption was 363.258 billion yuan, a 40.8% increase in year-on-year terms. Third, the supporting provinces to Xinjiang have given full play to the resource advantages of talent and intelligence in guiding and helping agricultural and livestock products to improve quality and efficiency. They also help increase the support in production, processing, market sales, and brand building, and broaden the sales channels of agricultural and sideline products. Fourth, Big Data have been widely used in precision production and decision-making, food safety supervision, precision consumption marketing, and market and trade guidance [38]. *Note*: Data source: Xinjiang Statistical Yearbook 2018.

**Government.** In the past decade, for a healthy and rapid development of the meat industry, the state and autonomous regions have issued more than 15 supporting policies for the development of animal husbandry including the mutton industry, such as several proposals of the state council on promoting sound and rapid development of pastoral areas (2011), the construction plan for the comprehensive production capacity of 10 million new slaughter mutton sheep in Xinjiang for 2012–2015 (2012), and several opinions on promoting the high-quality development of animal husbandry (2020).

In 2017, the total investment in fixed assets of animal husbandry (fixed assets in the whole country by sector) was 13.655 billion yuan, an increase of 620% compared with 1.896 billion yuan in 2010. The number of sheep in stock increased from 30.1337 million in 2010 to 40.305 million in 2017 (an increase of 33.75%) and the mutton output increased from 469,000 tons in 2010 to 603,200 tons in 2019 (an increase of 28.6%). This provides a strong guarantee for the development of the Xinjiang mutton industry.

## Results and analysis

### Quantitative analysis results

The first component affecting the comprehensive competitiveness of the mutton industry is closely related to the total population, sheep raising scale, output value of the animal husbandry, and investment in agriculture, forestry, animal husbandry, and fishery, which reflects the basic investment situation of each province in the mentioned industries. The second component is closely related to mutton production, consumption levels of the residents, and sheep stud farms, which reflect the production in the mutton industry and the demand for mutton in the various provinces. The third component is mainly related to animal husbandry technical

institutions and technicians, which reflect the technical level of the mutton industry in the provinces. The fourth component is mostly related to the consumption levels of residents and the investment in agriculture, forestry, animal husbandry, and fishery, which reflects the competitiveness of the related indicators of the mutton industry that are not included in the first three principal components. The correlation analysis and principal component analysis results showed that the mutton production in 2017 was among the top 10 provinces in China. Inner Mongolia tops in terms of the comprehensive competitiveness of the mutton industry, followed by Xinjiang, and lastly, Henan.

## Qualitative analysis results

The natural resources of production factors are relatively rich, and the advanced production factors are large but not strong; in terms of demand conditions, the natural feeding conditions of Xinjiang sheep, the special quality of mutton and a unique practice of mutton-cooked products have established a special reputation for Xinjiang mutton. For the performance of related and supporting industries, planting industry provides a guarantee, but the industrialization level of other supporting industries is low; meanwhile, in terms of strategy, structure, and competitors, enterprises do not have advantages. The development of "belt and road" and the policy of aiding Xinjiang and the tourism industry will provide a good opportunity for developing the mutton industry. The government's auxiliary elements are in general.

## Conclusion

This study uses Michael Porter's "diamond model" to analyze the competitiveness of Xinjiang's mutton industry quantitatively and qualitatively. Xinjiang ranked second in the mutton production all over China. Xinjiang mutton industry has no competitive advantage in four main factors: production factors, related and supporting industries, enterprise strategy, and competitors. It has a competitive advantage in the main elements of demand conditions and auxiliary elements of opportunities. Moreover, Xinjiang has become the second-largest province of mutton production and the largest province of mutton consumption per capita. In recent years, the output, price, and demand of mutton in Xinjiang have maintained the growth trend, and its new consumption demand is met by mutton imported to Xinjiang. With the steady growth of mutton demand in China, "belt and road" continue to develop. Moreover, with the continuous development and the rapid growth of Xinjiang's aid work and tourism, there is still room for the development of Xinjiang's mutton. Consequently, Xinjiang's high-quality mutton products will be favored by consumers.

## The deficiency of the research and the future research direction

In this paper, the position of Xinjiang mutton industry competitiveness, the mutton production in 2017 in the country's top 10 provinces as the comparative object, reflects the overall situation of Xinjiang mutton industry. Due to the limitations of time, energy, and objective factors, this study still has the following deficiencies and problems: First, this paper did not compare the competitiveness of the Xinjiang mutton industry with that of foreign mutton import and export regions. Second, the competitiveness of free-range mode and small slaughterhouses does not undergo a comparative analysis. Third, this paper does not adopt the "opportunity" auxiliary factor in quantitative analysis because of the difficulty in quantifying this factor. Fourth, the competitiveness index system of the mutton industry does not directly reflect the scale level, and the index system needs to be further improved. In addition, due to the small proportion of mutton industry in agriculture, data related to the development of the mutton industry are lacking. This lack of key data seriously limits further research of this

paper, thus making it difficult to draw relevant research conclusions and conduct empirical comparative analysis using detailed data or rigorous econometric model. However, with the implementation of national support policies for the mutton industry and the development of specialization and scale of mutton industry, the relevant data of industrial development will continue to be complete and systematic. In the future, we can conduct a more in-depth empirical analysis on the competitiveness of mutton industry in Xinjiang.

## Enlightenment

**Rational allocation of production factors.** We should promote the adjustment of breed structure, actively introduce multiple breeds, crossbreed with local breeds, cultivate multiple breeds suitable for the local production environment, improve lambing rate, increase mutton production, and effectively solve the new demand for mutton. To tackle these problems, the government should further provide preferential policies to scientific research institutions and enterprises to introduce talents and basic research and development, increase the investment in scientific research and reward funds, and further promote the transformation of scientific and technological achievements.

**Increasing the support for related industries.** The government should introduce preferential policies to cultivate further and support the brand enterprises and the cold-chain logistics enterprises, promote specialization in the mutton industry, and create a low-cost transportation environment for livestock products. Besides, there is a need to create a Big Data platform, apply the technology to management and operations, and transform the traditional industries into smart ones.

**Optimize the supply side.** For the whole country, the mutton market space is large, which is conducive to the development of the mutton industry. Therefore, the levels of large-scale and intensive breeding, the quality of mutton products and the regional mutton brand awareness in Xinjiang must be improved, and differentiated production must be increased to meet the local market demand and expand it to the national market.

**Optimizing the development strategy.** Xinjiang mutton origin factors will bring differences in mutton brand image. Therefore, the image of geographical indications must be enhanced. Moreover, the brand association of Xinjiang mutton products must be stimulated among consumers. Moreover, the mutton market must be subdivided, and brand positioning must be carried out according to the preferences and the consumption habits of the different consumers; thus, different brands are precisely designed for different consumer groups. Lastly, to establish a national brand-strategic plan, we must draw lessons from the "national brand" planning of China Central Television (CCTV), establish and improve the popularity of the local characteristics of the brands, and create high-grade mutton brands with these characteristics.

**Seize opportunities and promote development.** The opportunities that are derived from the "belt and road" initiative, the Xinjiang delicacy fever and Xinjiang food, and the Big Data application development should be fully utilized. We should seize the leading role of tourism development and inter-provincial aids to Xinjiang provinces and cities, actively open up the product marketing channels, further uncover and give full space to the food resources of "Xinjiang flavor," and improve the recognition of Xinjiang mutton brand in the coastal markets. To establish mutton-catering enterprises with regional characteristics to drive the development of the mutton industry, enterprises can also draw experience from other catering enterprises in Inner Mongolia, such as "Small sheep" and "Wellamb."

**Increasing the policy support and guidance from the government.** The government should consider the mutton industry as the main measure to eliminate poverty and rural

vitalization, introduce relevant supporting policies, further increase the support in policy and finance, provide a good policy environment, and achieve accurate support. We should further coordinate and solve the contradictions between the "visible hand" and the "invisible hand," take good advantage of the support points of the government, establish and improve a good financing environment, and eliminate government-led capital investment. Moreover, governmental efforts to purchase public services must be increased, given the ample leading space to aquaculture enterprises and large farmers in the market. Further, we must provide adequate room to the complementary role of the "two hands" (the government and market) and finally maximize the efficiency.

## Author Contributions

**Conceptualization:** Ablat Dawut, Yingjie Tian.

**Data curation:** Ablat Dawut.

**Formal analysis:** Ablat Dawut.

**Investigation:** Yingjie Tian.

**Methodology:** Ablat Dawut, Yingjie Tian.

**Software:** Ablat Dawut.

**Visualization:** Ablat Dawut.

**Writing – original draft:** Ablat Dawut, Yingjie Tian.

**Writing – review & editing:** Ablat Dawut, Yingjie Tian.

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
