## [Decision Letter · Decision Letter 0]

20 Apr 2021

PONE-D-21-00814

Research on the Competitiveness of the Xinjiang Mutton Industry based on Diamond Model

PLOS ONE

Dear Dr. Ablat,

Thank you for submitting your manuscript to PLOS ONE. After careful consideration, we feel that it has merit but does not fully meet PLOS ONE’s publication criteria as it currently stands. Therefore, we invite you to submit a revised version of the manuscript that addresses the points raised during the review process.

We look forward to receiving your revised manuscript.

Kind regards,

Bing Xue, Ph.D.

Academic Editor

PLOS ONE

Journal Requirements:

4. Please ensure that you refer to Figure 6 in your text as, if accepted, production will need this reference to link the reader to the figure.

Reviewers' comments:

Reviewer's Responses to Questions

**Comments to the Author**

1. Is the manuscript technically sound, and do the data support the conclusions?

Reviewer #1: Yes

Reviewer #2: Partly

2. Has the statistical analysis been performed appropriately and rigorously? 

Reviewer #1: No

Reviewer #2: No

3. Have the authors made all data underlying the findings in their manuscript fully available?

Reviewer #1: Yes

Reviewer #2: Yes

4. Is the manuscript presented in an intelligible fashion and written in standard English?

Reviewer #1: No

Reviewer #2: Yes

5. Review Comments to the Author

Reviewer #1: (1) Few documents cited, and the format of the introduction is not standardized;(2) Many articles are not cited in the introduction, but there are many in the method, which is a bit reversed;(3) There is no in-depth discussion and analysis of this article;(4)The method is too simple and the description is too little;(5)Correlation analysis cannot explain the problem in detail. Correlation coefficient and p-value are needed to see the accuracy of calculation.

Reviewer #2: 1.what are the reasons for the lack of competitiveness of the Xinjiang mutton industry? what are your suggestions for improving competitiveness? These reasons and suggestions should be clarified briefly in the abstract so that the readers can know them quickly.

2. why you choose the diamond model to analyze the Xinjiang mutton industy? I think the author should give some reasons before using this model.

3. Line 126-128 are confusing.

4. index selection process is not clear enough. I hope the author can mkae it clear so that other researchers can follow this selction method.

5. Line 376-399 list so many supporting government policies. I think it is not necessary. it is more important to analyze the supporting effects of these policies.

6. Line 449-454 are not the results of this paper.

7. the deficiency of the research and the future research direction part should be rewritten. the contents of this part are not consist with the title.

6. PLOS authors have the option to publish the peer review history of their article (what does this mean?). If published, this will include your full peer review and any attached files.

Reviewer #1: No

Reviewer #2: No

---

## [Author Response · Author response to Decision Letter 0]

18 Jul 2021

Response to reviewers’ comments on PONE-D-21-00814

Dear editor of Plos One:

Thank you very much for your detailed evaluations and suggestions!

Your comments were highly insightful and enabled us to greatly improve the quality of our paper. We have carefully revised our manuscript entitled by "Competitiveness of Xinjiang’s Mutton Industry based on Diamond Model" according to your advice. Additionally, the editorial errors throughout the manuscript have been reedited and rectified. This manuscript was proofed by native speaker. We wish the revision could meet with your approval.

Here, we listed the point-by-point responses to your detailed comments and suggestions (With blue). As follows:

Journal Requirements:

1.Line 2: Please ensure that your manuscript meets PLOS ONE's style requirements, including those for file naming.

Response: I have changed the style of this paper according to the PLOS ONE's style.

2.Line 28: We suggest you thoroughly copyedit your manuscript for language usage, spelling, and grammar. If you do not know anyone who can help you do this, you may wish to consider employing a professional scientific editing service.

Response: Foreign editors have made a comprehensive language editing of this paper. See in supplementary information file (English editing by native speaker).

3.Line 29: PLOS requires an ORCID iD for the corresponding author in Editorial Manager on papers submitted after December 6th, 2016. Please ensure that you have an ORCID iD and that it is validated in Editorial Manager.

Response: I have applied ORCID iD, ORCID identifier is 0000-0002-6134-6107.

4.Line 300: Please ensure that you refer to Figure 6 in your text as, if accepted, production will need this reference to link the reader to the figure.

Response: I have added the name of (Figure 6) in this place.

Comments to the Author:

1.Line 55: Is the manuscript technically sound, and do the data support the conclusions?

Response: Yes

2.Line 239: Has the statistical analysis been performed appropriately and rigorously?

Response: The statistical data of this paper comes from China Statistical Yearbook and Xinjiang statistical yearbook. The data is reliable and the linear analysis method is used for statistical analysis. We use the package of SciPy in Python to check the significance of the Pearson correlation coefficient.

3.Line 269: Have the authors made all data underlying the findings in their manuscript fully available?

Response: Yes

4.Line 271: Is the manuscript presented in an intelligible fashion and written in standard English?

Response: In order to meet the requirements of the journal in english writing, this paper is revised thoroughly by Foreign editors. See in supplementary information file (English editing by native speaker).

5.Review Comments to the Author

Reviewer #1:

(1)Line 292: Few documents cited, and the format of the introduction is not standardized.

Response: According to the suggestions, the number of references is increased from 31 to 38. It is modified according to the PLoS One format.

(2)Line 308: Many articles are not cited in the introduction, but there are many in the method, which is a bit reversed.

Response: Seven references are added in the introduction. Indeed, we have cited many references in the method, mainly because we want to prove that the method of "Diamond Model" has reference basis.

(3)Line315: There is no in-depth discussion and analysis of this article.

Response: In the part of quantitative analysis, we made a comprehensive explanation of the evaluation index. In the process of selecting the evaluation index, we fully considered the scientificity and operability of the index, and achieved the comparability, rationality and objectivity of the results. In terms of tools, we have changed the original analysis tools. Now we use the package of SciPy in Python to check the significance of Pearson correlation coefficient one by one, and use the hypothesis method for correlation analysis.

(4)Line 356: The method is too simple and the description is too little.

Response: To solve this problem, we have modified the relevant content.

(5)Line 237: Correlation analysis cannot explain the problem in detail. Correlation coefficient and p-value are needed to see the accuracy of calculation.

Response: We use the package of SciPy in Python to check the significance of Pearson correlation coefficient one by one.

Reviewer #2:

(1)Line 94: what are the reasons for the lack of competitiveness of the Xinjiang mutton industry? what are your suggestions for improving competitiveness? These reasons and suggestions should be clarified briefly in the abstract so that the readers can know them quickly.

Response: In view of the above problems, some modifications have been made to the abstract.

(2)Line 107: Why you choose the diamond model to analyze the Xinjiang mutton industy? I think the author should give some reasons before using this model.

Response: In view of the above problems, we have made a detailed description in the reference and method part.

(3)Line 130: are confusing.

Response: I have changed (were transferred from other provinces) to (were transferred from Xinjiang to other provinces).

(4)Line 316: Index selection process is not clear enough. I hope the author can mkae it clear so that other researchers can follow this selction method.

Response: In the part of quantitative analysis, we have made a comprehensive explanation of the selection of evaluation indicators. In the process of selecting evaluation indicators, we have fully considered the scientificity and operability of the indicators, and achieved the comparability, rationality and objectivity of the results.

(5)Line 430-435 list so many supporting government policies. I think it is not necessary. it is more important to analyze the supporting effects of these policies.

Response: In this paragraph, we delete some document names and add fixed asset investment content and mutton production to further illustrate the support effect of the policy.

(6)Line 479-497 are not the results of this paper.

Response: This content has been re-modified.

(7)Line 478: The deficiency of the research and the future research direction part should be rewritten. the contents of this part are not consist with the title.

Response: This content has been rewritten.

6.Line 512: PLOS authors have the option to publish the peer review history of their article (what does this mean?). If published, this will include your full peer review and any attached files.

Response: Yes

7.Line 527: Do you want your identity to be public for this peer review? For information about this choice, including consent withdrawal, please see our Privacy Policy.

Response: No

Other modifications:

1.Line 147-148: I have new added this part.

2.Line 148-149: This content is moved here from the first page.

3.Line 149-153: I have new added this part.

4.Line 153-157: This content is moved here from the first page.

5.Line 194-195: I have new added this part.

6.Line 207-215:I have new added this part.

7.Line 216: I have deleted the word (Index selection) from this place.

8.Line 513: In this place, i have deleted (Improving demand conditions), and added (Optimize the supply side).

9.Line 515: I have new added this.

---

## [Decision Letter · Decision Letter 1]

8 Sep 2021

Competitiveness of Xinjiang’s Mutton Industry based on Diamond Model

PONE-D-21-00814R1

Dear Dr. Dawut,

We’re pleased to inform you that your manuscript has been judged scientifically suitable for publication and will be formally accepted for publication once it meets all outstanding technical requirements.

Kind regards,

Bing Xue, Ph.D.

Academic Editor

PLOS ONE

Additional Editor Comments (optional):

Reviewers' comments:

Reviewer's Responses to Questions

**Comments to the Author**

1. If the authors have adequately addressed your comments raised in a previous round of review and you feel that this manuscript is now acceptable for publication, you may indicate that here to bypass the “Comments to the Author” section, enter your conflict of interest statement in the “Confidential to Editor” section, and submit your "Accept" recommendation.

Reviewer #1: All comments have been addressed

Reviewer #2: All comments have been addressed

2. Is the manuscript technically sound, and do the data support the conclusions?

Reviewer #1: Yes

Reviewer #2: Yes

3. Has the statistical analysis been performed appropriately and rigorously? 

Reviewer #1: Yes

Reviewer #2: Yes

4. Have the authors made all data underlying the findings in their manuscript fully available?

Reviewer #1: Yes

Reviewer #2: Yes

5. Is the manuscript presented in an intelligible fashion and written in standard English?

Reviewer #1: Yes

Reviewer #2: Yes

6. Review Comments to the Author

Reviewer #1: Clear thinking, rigorous discussion process, reasonable analysis, the result is strong in practical application, but the language can continue to be strengthened

Reviewer #2: I have noticed that most of my comments had been incoprorted in this revised version. Data and analysis are detailed and convincing. I have learnt from the authors.

7. PLOS authors have the option to publish the peer review history of their article (what does this mean?). If published, this will include your full peer review and any attached files.

Reviewer #1: No

Reviewer #2: No

---

## [Editor Report · Acceptance letter]

13 Sep 2021

PONE-D-21-00814R1 

Competitiveness of Xinjiang’s Mutton Industry based on Diamond Model 

Dear Dr. Dawut:

I'm pleased to inform you that your manuscript has been deemed suitable for publication in PLOS ONE. Congratulations! Your manuscript is now with our production department. 

Kind regards, 

on behalf of

Professor Bing Xue 

Academic Editor

PLOS ONE